# The Effects of Silver Sulfadiazine on Methicillin-Resistant *Staphylococcus aureus* Biofilms

**DOI:** 10.3390/microorganisms8101551

**Published:** 2020-10-08

**Authors:** Yutaka Ueda, Motoyasu Miyazaki, Kota Mashima, Satoshi Takagi, Shuuji Hara, Hidetoshi Kamimura, Shiro Jimi

**Affiliations:** 1Department of Pharmacy, Fukuoka University Hospital, Fukuoka 814-0180, Japan; ueday@fukuoka-u.ac.jp (Y.U.); mashimakota210@fukuoka-u.ac.jp (K.M.); kamisan@fukuoka-u.ac.jp (H.K.); 2Department of Pharmacy, Fukuoka University Chikushi Hospital, Fukuoka 818-8502, Japan; motoyasu@fukuoka-u.ac.jp; 3Departments of Plastic, Reconstructive and Aesthetic Surgery, Faculty of Medicine, Fukuoka University, Fukuoka 814-0180, Japan; stakagi@fukuoka-u.ac.jp; 4Department of Drug Informatics, Faculty of Pharmaceutical Sciences, Fukuoka University, Fukuoka 814-0180, Japan; harashu@fukuoka-u.ac.jp; 5Central Lab for Pathology and Morphology, Faculty of Medicine, Fukuoka University, Fukuoka 814-0180, Japan

**Keywords:** biofilm, MRSA, silver ion, silver sulfadiazine, wound infections

## Abstract

Methicillin-resistant *Staphylococcus aureus* (MRSA), the most commonly detected drug-resistant microbe in hospitals, adheres to substrates and forms biofilms that are resistant to immunological responses and antimicrobial drugs. Currently, there is a need to develop alternative approaches for treating infections caused by biofilms to prevent delays in wound healing. Silver has long been used as a disinfectant, which is non-specific and has relatively low cytotoxicity. Silver sulfadiazine (SSD) is a chemical complex clinically used for the prevention of wound infections after injury. However, its effects on biofilms are still unclear. In this study, we aimed to analyze the mechanisms underlying SSD action on biofilms formed by MRSA. The antibacterial effects of SSD were a result of silver ions and not sulfadiazine. Ionized silver from SSD in culture media was lower than that from silver nitrate; however, SSD, rather than silver nitrate, eradicated mature biofilms by bacterial killing. In SSD, sulfadiazine selectively bound to biofilms, and silver ions were then liberated. Consequently, the addition of an ion-chelator reduced the bactericidal effects of SSD on biofilms. These results indicate that SSD is an effective compound for the eradication of biofilms; thus, SSD should be used for the removal of biofilms formed on wounds.

## 1. Introduction

Biofilms (BFs) are a cause of chronic infections. Several types of symbiotic bacteria, such as *Staphylococcus aureus* and *Pseudomonas aeruginosa*, colonize our body and form BFs [1,2,3]. Methicillin-resistant *S. aureus* (MRSA) causes soft-tissue infections, indwelling catheter-associated infections, bacteremia, endocarditis, and osteomyelitis. Approximately 80% of chronic wound infections are attributed to bacteria or BFs [1]. BFs produce a subpopulation of drug-resistant cells called persister cells [4,5,6,7]. The BF matrix predominantly contains extracellular polysaccharides [8,9], and interacts with other molecules, including quorum-sensing signaling molecules/autoinducers, polypeptides, lectins, lipids, and extracellular DNA [10,11,12]. Specific molecules targeting BFs and effective drugs for BF eradication have not been identified yet. Therefore, the BF itself becomes a serious exacerbation factor in antimicrobial resistance.

Silver sulfadiazine (SSD) has been used as an exogenous antimicrobial agent since the 1960s [13,14], and is used on partial and full thickness burns to prevent infection [15,16]. It is registered on the World Health Organization’s List of Essential Medicines [17]. SSD possesses broad-spectrum antibacterial activity, reacting nonspecifically to Gram-negative and Gram-positive bacteria, causing distortion of the cell membrane and inhibition of DNA replication [14,18,19,20]. Nevertheless, common side effects of SSD include pruritus and pain at the site of administration [21], as well as decreased white blood cell counts, allergic reactions, bluish-gray skin discoloration, and liver inflammation [22,23,24]. However, the Cochrane systematic review (2010) [25] did not recommend the use of SSD because of insufficient evidence on whether silver-containing dressings or topical agents promote wound healing or prevent wound infection. In addition, specific antibiotics are currently obtainable.

To overcome these shortcomings, new nanomedicine technologies using SSD have been adopted to enhance its antibacterial activity [26]. Unfortunately, SSD cytotoxicity was also enhanced [27]. SSD is poorly soluble and has limited penetration through intact skin [28,29]. When it comes in contact with body fluids, free sulfadiazine (SD) can be absorbed systemically, metabolized in the liver [30], and excreted in urine [31].

Notably, the effects of SSD on BFs have been widely studied [32,33]; however, their mechanisms of action have not been investigated yet. In this study, we investigated the essential mechanisms underlying the antibacterial action of SSD, especially against BFs, using SSD elements, silver and SD, and analyzed the effects of the compounds on a clinical strain of BF-forming MRSA.

## 2. Materials and Methods

### 2.1. Preparation of BF Chips

ATCC BAA-2856 (OJ-1) [34,35,36], a high-BF-forming strain of MRSA, was used. BF chips were prepared as previously described [36,37]. In brief, one colony grown on tryptic soy agar (TSA) was inoculated in tryptic soy broth (TSB) and grown at 37 °C until the optical density (OD) = 0.57 (λ = 578 nm). A 1000-fold diluted bacterial solution was used for BF formation in the culture, and a plastic sheet for an overhead projector (3M Japan Ltd., Tokyo, Japan) was used as the substrate. Sterilized plastic sheets (1 × 8 cm) were immersed in bacterial solution and incubated to obtain a mature and uniform BF. After incubation, the sheets were washed 3 times with 10 mL of 0.01 M phosphate buffered saline solution (pH 7.4) to remove planktonic cells. Plastic sheets cut into 1 × 1 cm pieces termed “BF chips” were primarily used in the experiments unless otherwise stated.

### 2.2. Determination of BF Mass and Number of Live Bacteria

BF mass using crystal violet (CV) stain and live bacteria number by colony forming unit (CFU) counts were determined using a BF sheet (1 × 8 cm) in 10 mL media. CV-positive BF mass was measured according to a previously reported method [38] with some modifications. In brief, BFs stained with CV (Sigma-Aldrich, Tokyo, Japan) were eluted in 1 mL of 30% acetic acid in an RIA tube (Eiken Chemical Co., Ltd., Tokyo, Japan). Absorbance (λ = 570 nm) was measured using a spectrometer (GENESYS 10S VIS, Thermo Scientific, LMS, Tokyo, Japan). Bacteria on BF sheets were dissociated using an ultrasonic generator (Sonifier 250; Branson Ultrasonics, Emerson Japan, Ltd., Kanagawa, Japan), and colony-forming units (CFUs) were assessed.

### 2.3. Antibacterial Effects of Compounds on BFs

Ethylene oxide-gas sterilized silver nitrate (FUJIFILM Wako Pure Chemical Corporation, Osaka, Japan), SSD (ALCARE Co., Ltd., Tokyo, Japan), and SD (FUJIFILM Wako Pure Chemical Corporation) were freshly prepared in TSB at a maximum concentration of 11,200 μM and serially diluted to 43.75 μM. After dipping the BF chip in 5-mL TSB-containing tubes in the presence of different concentrations of AgNO_3_, SSD, and SD, the tubes were incubated at 37 °C for 24 h. Because SSD and SD were insoluble in liquid, cultures were continuously and gently mixed on a horizontal-rotation shaker (G10: New Brunswick Scientific, New Brunswick, NJ, USA) at 120 rpm.

The minimum antibacterial concentrations of compounds for bacteria derived from BF were determined by methods described previously [37] with some modifications (Appendix A). After incubating the bacteria for 24 h with different compounds at different concentrations, the culture tubes were kept at 4 °C for 1 h in order for the compounds to sediment, before the turbidity in the supernatant was measured (λ = 578 nm). The values were used to assess the minimum inhibitory concentration for planktonic cells derived from BFs (bMIC). Next, 10 μL of medium from each previously used tube for the bMIC analysis, was blotted on a 1 × 1 cm filter paper on TSA, and then they were incubated overnight at 37 °C. The colonies that grew around the paper were used to assess the minimum bactericidal concentration for planktonic cells derived from BF (bMBC). BF chips used for the bMIC analysis were directly placed on TSA and incubated overnight at 37 °C, and the minimum BF eradication concentration (MBEC) was determined.

In some studies, ethylenediamine-*N*,*N*,*N′*,*N′*-tetraacetic acid (EDTA) (FUJIFILM Wako Pure Chemical Corporation) (pH 7.4) at a concentration of 680 μM was used to chelate ions including silver.

### 2.4. Measurement of Liberated Silver Ions in the Media

After BF chips were incubated in TSB at 37 °C for 24 h in the presence of different concentrations of AgNO_3_ and SSD, media were centrifuged (EX-136: TOMY SEIKO Co. Ltd., Tokyo, Japan) at 3000 rpm for 10 min, and filtered with a 0.45-μm filter unit (Merck Millipore, Darmstadt, Germany) to remove bacteria and SSD aggregates. Ionized silver in the media was diluted 4 times with water and was quantified with AGT-131 using a NI-Ag kit (range: 0.01–0.25 ppm Ag-ion/mg (mL), Japan Ion Co., Tokyo, Japan).

### 2.5. SSD Inducing Direct/Indirect Bactericidal Effects on BFs

To avoid direct contact of insoluble SSD with BFs, a closed chamber with a semipermeable membrane (Intercell S well chamber: KURABO INDUSTRIES LTD, Osaka, Japan) was used along with a syringe gasket to close the lid. Different concentrations of SSD in 500 μL TSB were injected into the chamber, which rested on 4.5 mL of TSB with the BF chip in a tube. The tube was cultured at 120 rpm at 37 °C for 24 h. For the control, 500 μL of SSD was directly added into TSB with the BF chip, and the chamber was filled with 500 μL TSB floated on the medium. After incubation, bMIC, bMBC, and MBEC were determined.

### 2.6. Quantification of the SD Attachment on BFs

The SD amount was quantified using the diazo-coupling reaction with *N*,*N*-diethyl-*N*′-1-naphthylethylenediamine oxalate, also known as Tsuda’s reagent (FUJIFILM Wako Pure Chemical Corporation) [39]. BFs formed on the surface of plastic tubes after 24 h incubation at 37 °C were washed 3 times with phosphate buffered saline. Freshly prepared SD was added at concentrations between 43.75 and 11,200 μM in 3 mL of TSB in a tube and was incubated at 37 °C for 1 h. After incubation, BFs were washed 3 times to remove any unbound SD on the BFs, and 500 μL of 1 N HCl was added to the tubes, from which 70 μL of the solution was mixed with 20 μL of 10% sodium nitrite (FUJIFILM Wako Pure Chemical Corporation) and reacted for 2 min on ice. Then, 100 μL of 2.5% ammonium amidosulfate (FUJIFILM Wako Pure Chemical Corporation) was added and reacted for 1 min, and 100 μL of Tsuda’s reagent was added to the solution and mixed. The solutions in a 96-well plate (Becton, Dickinson and Company, Franklin Lakes, NJ, USA) were measured (λ = 550 nm) using a microplate reader (Model680; Bio-Rad Laboratories, Inc., Hercules, CA, USA). The SD calibration curve was also prepared (Appendix A).

### 2.7. Morphological Analysis of the Effects of AgNO_3_, SSD, and SD on BFs

Plastic sheets in TSB with or without bacteria were incubated at 37 °C for 24 h. After incubation, the sheets were placed in media containing AgNO_3_, SSD, and SD at a concentration of 2800 μM and incubated at 37 °C for 24 h. After incubation, the sheets were fixed in 5% formalin (pH 7.4), and stained with crystal violet. Another set of unstained sheets was placed in the air for more than one week to be turned black due to the sulfur-oxidized silver properties.

### 2.8. Ethics Approval

All methods involving bacterial handling were performed in accordance with the relevant guidelines and regulations under the Fukuoka University’s experiment regulations.

### 2.9. Data and Statistical Analysis

Results from two different experimental groups initially underwent a distribution analysis using the F-test, before the Student’s *t*-test or Mann–Whitney *U* test were performed. *p* values  < 0.05 were considered to denote statistical significance. Sample numbers and repeated experiments are indicated in the legends of the figures and tables. Data are expressed as mean ± standard error.

## 3. Results

### 3.1. Antibacterial Effects of AgNO_3_, SSD, and SD

Antibacterial effects of AgNO_3_, SSD, and SD on BF chips were obtained (Table 1): bMIC for planktonic bacteria from BFs: 700 μM, 700 μM, >5600 μM; bMBC for planktonic bacteria from BFs 2800 μM, 1400 μM, >5600 μM; and MBEC for bacteria in BFs: >5600 μM, 2800 μM, >5600 μM, respectively. Results showed that SD had no antibacterial effects, but SSD alone was effective in targeting BFs.

### 3.2. Effects of AgNO_3_, SSD, and SD on Viable Cells in BFs

Viable cell numbers on the BF chip were analyzed (Figure 1) using a CFU assay technique after collecting all bacteria from the BF chip. After 24-h incubation with AgNO_3_ in different concentrations, bacterial growth was all arrested to some extent, but a dose-dependent suppression effect was not found. In contrast, SSD suppressed CFU values in a dose-dependent manner, especially at concentrations higher than 2800 μM, and the values reached approximately 1/100 of the initial density (0 h) and approximately 1/100,000 of cell density after 24 h of incubation. However, no effects on cell growth were found in any of the SD exposures.

### 3.3. Silver Ion Release in the Culture with SSD

Antibacterial effects of SSD could be due to the ionized silver’s action. As such, silver ions liberated in the cultures with AgNO_3_ and SSD were measured. In both cases, liberated silver ions increased dose-dependently (Figure 2). Silver ions in the culture with AgNO_3_ demonstrated a linear increase pattern, while those in the culture with SSD showed a logarithmic increase pattern. Significant differences were noted in the samples with concentrations greater than 700 μM (Figure 2). In the cultures with AgNO_3_ and SSD at the concentration of 700 μM, silver ions increased to 5.91 μM and 4.64 μM, respectively. At the maximum concentration (11,200 μM), the silver ion concentration in the culture with AgNO_3_ was three times higher than that in SSD culture (Figure 2).

### 3.4. Direct/Indirect Bactericidal Effect of SSDs on BFs

We hypothesized that the antibacterial effects of SSD could be due to its direct attachment to the BFs. To examine this mechanism, SSD was confined to a chamber, and a similar culture study was carried out. The bMIC in the culture with the chambered SSD was 1400 μM, which was two times greater than that in the culture with direct addition of SSD (Table 2). In terms of the bactericidal effects, the bMBC increased to >5600 μM, which was greater in the chambered SSD condition compared to that by direct SSD addition. Similarly, the MBEC value also increased to >5600 μM, which was greater in the chambered SSD compared to that in the direct SSD condition (Table 2). Therefore, the direct attachment of SSD to the BFs was necessary for initiating its antibacterial effects.

### 3.5. Effects of EDTA on AgNO_3_ and SSD-Induced Antibacterial Activity

EDTA is a chelator for silver ions [40]. To analyze the involvement of silver ions in the antibacterial effects, EDTA at a concentration of 640 μM was used. In the cultures with AgNO_3_, EDTA did not affect bMIC, bMBC, and MBEC values (700 μM, 2800 μM, and >5600 μM, respectively, Table 2). However, all the values in the cultures with SSD were increased by the addition of EDTA (700 μM → 1400 μM, 1400 μM → 2800 μM, and 2800 μM → >5600 μM, respectively, Table 2).

### 3.6. Effects of Compounds on BF Eradication

The BF eradication effects of the compounds were analyzed at a dose of 2800 μM, an effective concentration for SSD (Table 2). The results of BF mass, CFU and morphological analysis are shown in Figure 3a–c, respectively. The BF eradication effects were shown to be SSD > AgNO_3_ > SD, and SD had no effect. Silver-containing compounds, including AgNO_3_ and SSD, displayed BF eradication activities, where SSD showed the greatest effects.

Given that SSD is chemically synthesized with SD and silver, we examined the BF eradication effects of a mixture of SD and AgNO_3_ at a concentration of 2800 μM (Figure 3a,c). The values of BF mass and live bacteria number in the mixture were lower than those in SD alone, equivalent to those in AgNO_3_ alone, but were still significantly higher than those in SSD, suggesting that SSD was still the most effective at eradicating BFs.

### 3.7. SD Deposition to BFs

In the control without BFs, the addition of any concentration of SD to the media only resulted in a consistently low level of SD being deposited on the plastic surface (Figure 3d). However, in the BF tubes, SD was deposited on the BFs in a dose-dependent manner, and prominent deposition started once concentrations of SD were greater than 1400 μM (Figure 3d).

### 3.8. BF Morphology

After exposure to the different compounds, the BFs stained with crystal violet are shown in the left panels of Figure 3b, where the overall purple staining intensity among the groups follows the next scheme: Control > SD > AgNO_3_ > SSD. String-like structures (i.e., thickened BFs) found in the Control group remained in the SD and AgNO_3_ groups, and such structures were scarcely detected in the SSD group.

### 3.9. Localization of Silver on BFs

Oxidized silver turns black in color. No black spots were found on the plastic substrate even after exposure to all compounds, nor were there spots on the BFs following SD exposure (Figure 3b, right panels). In contrast, fine small black spots accumulated on the BFs after exposure to AgNO_3_ and SSD. Moreover, SSD induced denser accumulation of black spots.

## 4. Discussion

BFs are formed by bacteria, which contain massive extracellular polysaccharides/exopolymeric substances, and acquire an antibiotic-tolerant nature, which is explained by the decreased drug penetration [41] and the appearance of dormant cells (i.e., persister cells) [42]. Thus, the eradication of BFs has become a difficult task. In this study, we examined the effect of SSD on MRSA-formed BFs. The SSD concentration in the medical drugs, namely Silvadene, etc., is of 1% (10,000 μg/mL), which could be diluted with tissue fluid exuded from wounds after application. The highest concentration used in this study was 4000 μg/mL (0.4%: 11,200 μM), which is slightly lower than that of the medical drug.

The bactericidal actions of silver can be explained through three different mechanisms: (1) the production of dissolved oxygen-derived reactive oxygen species by its catalytic activity [43,44]; (2) the cross-linkage with silver at the sites of hydrogen bonding between the double strands of DNA [44,45]; and (3) the inhibition of enzyme activities by intracellular silver ions [44,46]. SD, a sulfonamide, inhibits intracellular folate metabolism in bacteria, resulting in proliferation arrest. During the investigation of silver-containing agents, SD has been selected amongst different compounds due to its high and broad bactericidal effects [14,18,19,20]. We showed here that SSD is effective against MRSA, especially in the BF state, but SD itself had no effect.

Among the compounds tested, SSD significantly decreased BF mass and live bacteria number in BFs, which was a considerably greater difference than that of AgNO_3_. This phenomenon was supported by silver deposition on the BF chip. We also examined the efficacy of the simultaneous addition of AgNO_3_ and SD instead of the SSD. These effects are equivalent to those of AgNO_3_ alone, but are significantly lower than SSD, indicating that the molecular form of SSD is important to induce a silver-related BF eradication. In contrast, it was reported that SSD was ineffective for MRSA BFs formed on a polycarbonate filter using a novel in vitro model (colony/drip-flow reactor) [32]. However, its exposure time was quite short (15 min) to induce an eradication reaction as compared with the present study (24 h).

In contrast, SSD and AgNO_3_ induced a similar antibacterial potency in bMIC and bMBC, both of which were detected in the planktonic state derived from the BFs. In accordance with their threshold levels, silver ion concentrations reached more than 5 μM. In any case, such concentration could be necessary for growth inhibition and the killing of planktonic bacteria. However, in the range of effective doses, silver ions were always generated at higher levels in AgNO_3_ conditions compared to those in SSD conditions. This reflects the contradiction of the bactericidal effects of SSD. The results suggest that some factor(s) other than simple diffusion of silver ion may be involved.

The MBEC level is a threshold for bacterial killing concentration in BFs. It was detectable only in SSD rather than AgNO_3_ or SD, which was also confirmed by the CFU assay. It clearly showed that AgNO_3_ could not efficiently kill the bacteria in BFs, and its effect remained within the level of growth inhibition. On the other hand, SSD strongly and dose-dependently depressed the live cell number in matured BFs, in which the live cell density at a concentration of 700 μM (the equivalent level of bMIC) reached 1/10,000 of the control after 24 h of incubation, and, at a concentration of 2800 μM (the equivalent level of MBEC), the number was 1/300 of the initial cell density and 1/300,000 of the control after 24-h incubation. These results suggest that the medial silver ion concentration is not a direct influencing factor for the SSD.

To clarify its mechanism of action, SSD was confined in a sealed chamber, by which SSD was constrained and bacteria could not penetrate beyond the membrane, but ionized silver could pass freely. As a result, silver ion concentrations in the chambered condition were lower at 700 and 1400 μM SSD, as compared with non-chambered conditions, but, in higher concentrations more than 2800 μM SSD, they became similar levels (Appendix A). Upon confinement, SSD-induced bMBC and MBEC levels were no longer seen. Our study also showed that SD itself had the property to bind to the BFs. Although, the exact binding mechanisms of SSD on BFs are still unknown, the direct attachment of SSD to BFs is crucial to induce bactericidal effects. Therefore, if the SSD binding site on BFs is identified in the near future, this would lead to the discovery of target molecules of the BFs formed by MRSA.

To validate the silver ion’s role in the antibacterial properties of SSD, EDTA was used at a concentration of 680 μM, which was about 40 to 100 times greater than that of the liberated silver ions in the media with AgNO_3_ and SSD. With the addition of EDTA, no changes in bMIC, bMBC, and MBEC were found in the culture with AgNO_3_. In contrast, the addition of EDTA increased in all of the values in the culture with SSD. The mechanisms of antibacterial action between AgNO_3_ and SSD are unknown; however, their mode of reaction might be different. Silver ions from AgNO_3_ could be easily bound by negatively charged components in media as compared to SSD, and a bound–liberation cycle may be repeated. Moreover, the silver holding capacity in SSD may be greater than that in AgNO_3_, by which SSD could reach the BFs. This mechanism may act especially on the BFs due to the selective adhesion of SD on BFs, which was confirmed by the greater deposition of silver on the BF chip incubated with SSD, whereas sites of the deposition on BFs could not be identified. It therefore seems likely that SSD may act on the bacteria settled in the BFs. After deposition of SSD, silver ions could be released in a micro environment in BFs, by which the opportunities for bacteria to kill could be increased, resulting in severe distortions of BF structure (Figure 3b).

As a limitation of the present study, one MRSA strain was only used. In future study, we will use different clinical strains, such as low-BF formers and high-BF formers in our collection [47].

## 5. Conclusions

SSD is a reliable anti-bacterial agent, and thus has recently been used as a coating material for indwelling catheters [48,49]. However, the mechanisms of action of SSD on BFs are still unclear. This study is the first to elucidate the mechanisms behind the efficacy of SSD on BFs. Therefore, it is possible that SSD preferably binds to BFs, and then it releases silver ions, by which bacteria settled in the BFs under a micro-environment are killed (Figure 4). In the future, further molecular levels of investigation are needed to identify the SSD binding site on BFs. Additionally, a systematic investigation of the use of SSD in BF-infected wounds should be performed in clinical practices rather than for infection prevention.

## Figures and Tables

**Figure 1 microorganisms-08-01551-f001:**
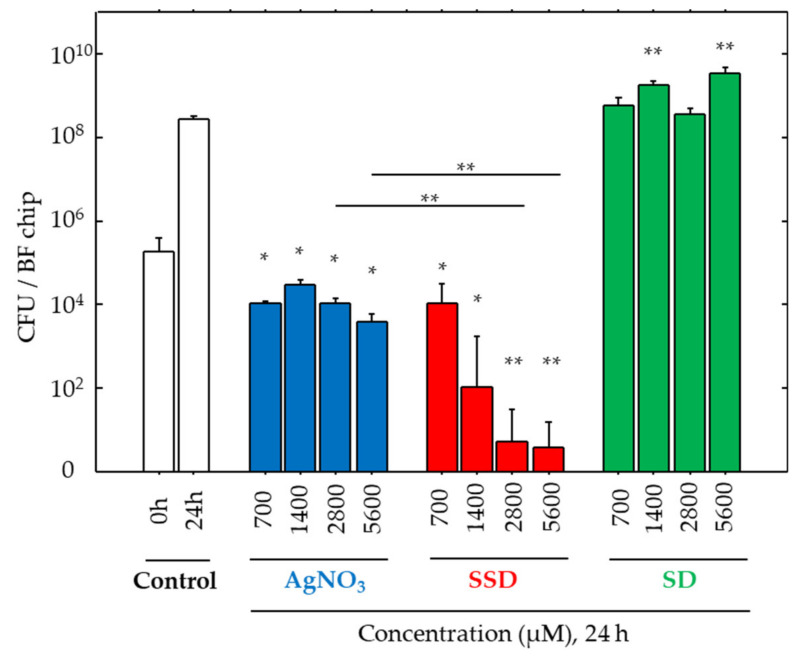
Colony forming unit (CFU) values for bacteria in biofilms (BFs). CFU values before (0 h) and after 24 h incubation at 37 °C in the vehicle alone (Control) and with AgNO_3_, silver sulfadiazine (SSD), and sulfadiazine (SD) at different concentrations. Data are presented as the mean ± SE. N size = 5; number of experimental replicates = 2. *: *p* < 0.05, **: *p* < 0.01 vs. control after 24-h incubation with different compounds.

**Figure 2 microorganisms-08-01551-f002:**
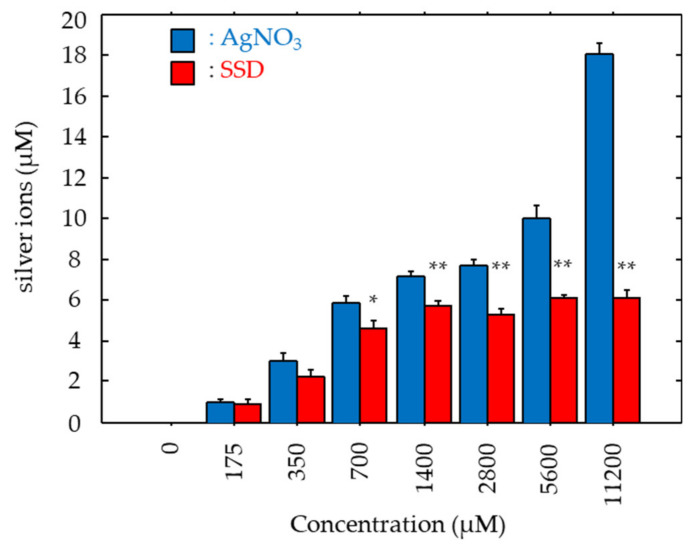
Medial ionized silver concentrations. After 24-h incubation at 37 °C in the presence of BF chips with AgNO_3_ and SSD, medial ionized silver concentration was determined. Data are presented as the mean ± SE. N size = 3; number of experimental replicates = 2. *: *p* < 0.05 and **: *p* < 0.01 vs. AgNO_3_.

**Figure 3 microorganisms-08-01551-f003:**
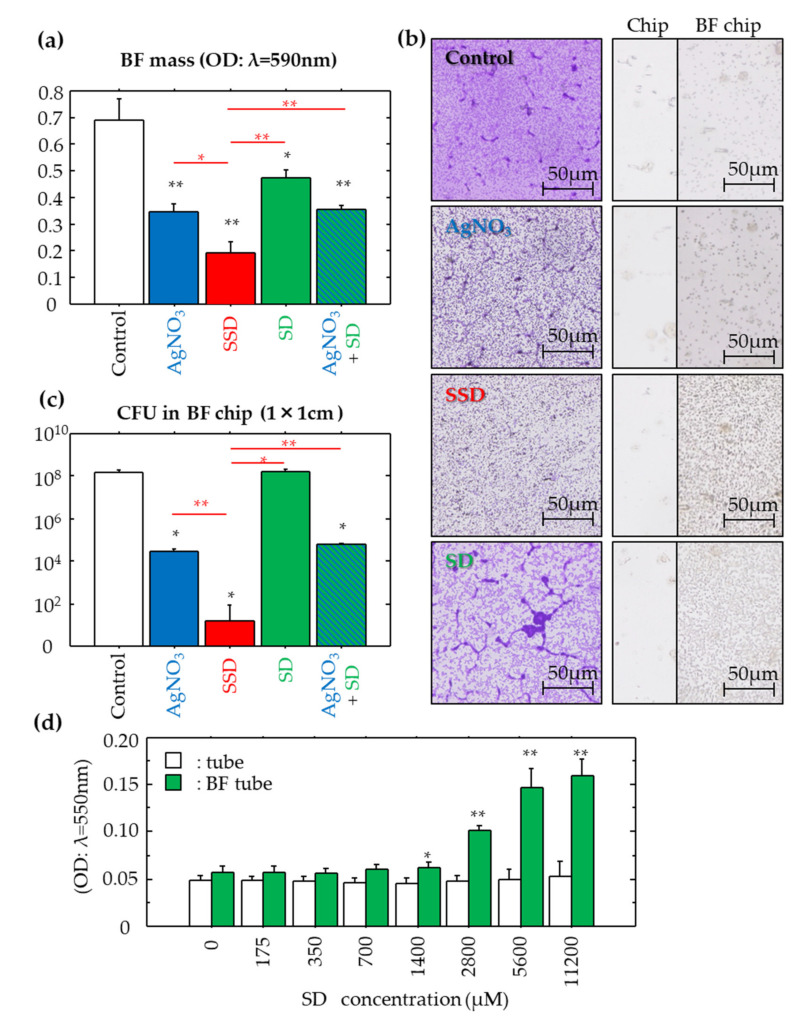
Effects of the compounds on mature BFs formed on plastic chips and SD deposition on BF chips. The effects of the compounds (2800 μM) after 24-h incubation at 37 °C on BF masses (**a**), morphology (**b**) and CFU values (**c**) were determined. Data are presented as the mean ± SE. N size = 5; number of experimental replicates = 2. *: *p* < 0.05 and **: *p* < 0.01 vs. negative control shown in white, and vs. SSD shown in red. In morphology, silver was deposited on the plastic chips with/without matured BFs after 24 h incubation at 37 °C with/without the different compounds. Sulfurized silvers in air are shown as black elements. Objective lens: ×20. In SD quantification, SD deposited on the BFs formed in plastic tubes after 24 h incubation at 37 °C was determined according to the method described in the Materials and Methods (**d**). Data are presented as the mean ± SE. N size = 5; number of experimental replicates = 2. *: *p* < 0.05 and **: *p* < 0.01 vs. BF-free control.

**Figure 4 microorganisms-08-01551-f004:**
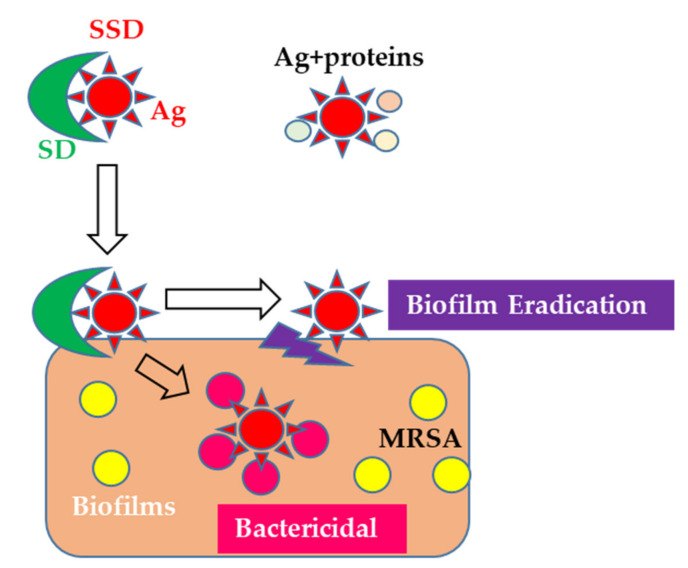
Working hypothesis of the action of SSD on biofilms formed by MRSA.

**Table 1 microorganisms-08-01551-t001:** Minimum antimicrobial concentration (bMIC) and minimum bactericidal concentration (bMBC) for planktonic bacteria from BFs and bacteria in BFs (minimum BF eradication concentration (MBEC)).

Compounds		0	175	350	700	1400	2800	5600 (μM)
	bMIC	+	+	+	−	−	−	−
AgNO_3_	bMBC	+	+	+	+	+/−	−	−
	MBEC	+	+	+	+	+	+	+
	bMIC	+	+	+	−	−	−	−
SSD	bMBC	+	+	+	−/+	−	−	−
	MBEC	+	+	+	+/−	−/+	−	−
	bMIC	+	+	+	+	+	+	+
SD	bMBC	+	+	+	+	+	+	+
	MBEC	+	+	+	+	+	+	+

The antimicrobial effects of different concentrations of the compounds were determined using the BF chip method described in the Materials and Methods. The results of independent replicated examinations were evaluated: “−”: 0/6 cases, “−/+”:2/6 cases, “+/−”:4/6 cases, “+”: 6/6 cases. N size = 6; number of experimental replicates = 2.

**Table 2 microorganisms-08-01551-t002:** Summary of antimicrobial threshold values.

	bMIC		bMBC		MBEC	
		(+EDTA)		(+EDTA)		(+EDTA)
<Direct>						
AgNO_3_ (μM)	700	(700)	2800	(2800)	>5600	(>5600)
SSD (μM)	700	(1400)	1400	(2800)	2800	(>5600)
SSD chambered (μM)	1400	(ND)	>5600	(ND)	>5600	(ND)

ND: not determined.

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
