# Peer review of "The Effects of Silver Sulfadiazine on Methicillin-Resistant Staphylococcus aureus Biofilms"

_microorganisms, 2020, doi:10.3390/microorganisms8101551_

Round 1

Reviewer 1 Report

Biofilms are a significant contributor to persistent infections and are difficult to treat. The authors aim to examine the mechanism of action of silver sulfadiazine (SDD) by analysis of the anti-microbial effects of its components separately and together. SDD has potent and broad-spectrum anti-bactericidal effects, but its applications are limited due to side-effects such as cytotoxicity. Use of an established biofilm test system with a single MRSA strain demonstrates that SDD is more effective than the individual active components and that it requires direct contact with the biofilm to mediate its effects. The authors suggest that further study of how SDD makes direct contact with the biofilm may be a useful direction for future work. This aim, methods and results in this manuscript are described clearly; and support the author’s conclusions. A limitation is that this study focuses on a single organism and strain.

Author Response

Thank you for your thoughtful review on our manuscript. As a limitation of this study you pointed-out, we have also awarded. In this study, to focus on the effects of SSD components on BF eradication from various aspects, we have used only one bacterial strain (ATCC BAA-2856). In future study, we are planning to analyze the SSD effects using different clinical MRSA strains, such as low-BF formers and high-BF formers in our collection (Toxins 2016;8:198). This information has been added at the end of Discussion. Therefore, please accept the current form regarding bacterial numbers.

Reviewer 2 Report

In this manuscript, using S. aureus, the authors studied the molecular mechanism of the anti-biofilm activity of silver sulfadiazine (SSD). As compared with AgNO3, SSD could kill S. aureus in biofilm, as measured by MBEC. The bactericidal effect of SSD was concentration-dependent. Although Ag ion was released from SSD, free Ag in the culture medium could not explain the antimicrobial and anti-biofilm activity of SSD. The anti-biofilm activity of the AgNO3 and SD mix was similar to that of AgNO3 and much lower than that of SSD. SD bound to S. aureus biofilm, and Ag deposition on biofilm was prominent with SSD. The authors proposed a model that SSD binds to the biofilm and release Ag ions, which kill bacteria in the biofilm. Based on these results, the authors also concluded that “SSD is an effective compound for the eradication of biofilms.”

Overall, the authors' conclusion is supported by the data presented. However, this manuscript has room for improvement. For example, some parts of this manuscript ( in particular, the results section) are hard to understand (see the comments below).

Major comment

Fig. S1: In the figure, bMIC was defined as the last chemical concentration in which bacterial growth is still evident (#5). I think it should be #6, the first tube where no apparent bacterial growth was observed.

Line 161: “Viable cell numbers on the BF chip were analyzed.”

: It would be helpful if the authors briefly describe how they did this experiment. As the current form, it is very difficult to figure out how the experiment was carried out.

Line 161 – 164: “In AgNO3, no changes in CFU value were found at any concentration as compared to the initial density on the BF chip prior to 24 h incubation.. 24 h incubation.”

: This part should be re-written. It is very confusing. The overall message seems that “ Although AgNO3 suppressed the growth of bacteria, the suppression was independent of concentration.”

Table 1 and Fig. 1: bMBC of AgNO3 is 2800 uM (Table 1). However, Fig. 1 showed that CFU was not significantly reduced by 2800 uM or above. In fact, AgNO3 concentration did not significantly affect the overall antibacterial activity of the compound. Why is that? Is this experiment different from the determination of bMBC in Table 1?

Line 180  - 181: “ .. the amount of silver ions released in cultures with AgNO3 increased about 3-times greater than that in SSD cultures.”

: I think the authors meant that “ …the silver concentration in the culture with AgNO3 was 3-times higher than that in SSD culture.”

Line 191 – 194: the bMBC increased to >5600 uM, which was over 4 times greater ….

                           Similarly, the MBEC value also increased to >5600 uM, which was over 2 times greater…”

: Technically the statement is correct. However,  >5600 uM means that the compound failed to kill all bacteria in medium (bMBC) or biofilm (MBEC) at the concentration.  I think the fold differences (e.g., 2 times, 4 times) do not have any quantitative meaning here.

Section 3.5: Does EDTA chelate silver?

Line 209 – 210: “The BF eradication effects among the various components were similar.”

: This sentence is not correct. SSD was most effective in eliminating BF.

Line 217: negative control shown in black.

: Negative control was shown in white.

Line 286 – 287: “As a result, silver ion concentrations in the media with or without the chamber were at similar levels (Figure S3).”

: This statement seems incorrect. Fig. S3b shows that the silver ion concentration in the chambered condition was lower at 700 and 1400 uM SSD, as compared with non-chambered conditions.

According to Ref # 32, SSD was not effective in killing MRSA in biofilm. I think this should be discussed.

Author Response

Q1: Fig. S1: In the figure, bMIC was defined as the last chemical concentration in which bacterial growth is still evident (#5). I think it should be #6, the first tube where no apparent bacterial growth was observed.

A1: Sorry, we have corrected.

Q2: Line 161: “Viable cell numbers on the BF chip were analyzed.”: It would be helpful if the authors briefly describe how they did this experiment. As the current form, it is very difficult to figure out how the experiment was carried out.

A2: We have added a brief method in the initial sentence.

Q3: Line 161 – 164: “In AgNO3, no changes in CFU value were found at any concentration as compared to the initial density on the BF chip prior to 24 h incubation.. 24 h incubation.”: This part should be re-written. It is very confusing. The overall message seems that “ Although AgNO3 suppressed the growth of bacteria, the suppression was independent of concentration.”

A3: Thank you for your suggestion. We have described more clearly in the first sentence as “bacterial growth was all arrested to some extent, but a dose-dependent suppression effect was not found”.

Q4: Table 1 and Fig. 1: bMBC of AgNO3 is 2800 uM (Table 1). However, Fig. 1 showed that CFU was not significantly reduced by 2800 uM or above. In fact, AgNO3 concentration did not significantly affect the overall antibacterial activity of the compound. Why is that? Is this experiment different from the determination of bMBC in Table 1?

A4: Your confusion may be derived from to our title in Table 1 and Figure 1. Therefore, we have changed them accordingly in the revised titles. To avoid such confusion, we have also added precise information in the text. We believed that they are clearer than before.

Q5: Line 180  - 181: “ .. the amount of silver ions released in cultures with AgNO3 increased about 3-times greater than that in SSD cultures.”: I think the authors meant that “ …the silver concentration in the culture with AgNO3 was 3-times higher than that in SSD culture.”

A5: Sorry about the point. We have revised it according to your suggestion.

Q6: Line 191 – 194: the bMBC increased to >5600 uM, which was over 4 times greater …. Similarly, the MBEC value also increased to >5600 uM, which was over 2 times greater…”: Technically the statement is correct. However,  >5600 uM means that the compound failed to kill all bacteria in medium (bMBC) or biofilm (MBEC) at the concentration.  I think the fold differences (e.g., 2 times, 4 times) do not have any quantitative meaning here.

A6: Sorry about the point. We have revised it according to your suggestion.

Q7: Section 3.5: Does EDTA chelate silver?

A7: At the top of the section, we have added a previous research as “EDTA is a chelator for silver ion” with a reference.

Q8: Line 209 – 210: “The BF eradication effects among the various components were similar.”: This sentence is not correct. SSD was s most effective in eliminating BF.

A8: The sentence was confused. We have revised it as “The BF eradication effects were shown to be SSD > AgNO3 > SD, and SD had no effect”.

Q9: Line 217: negative control shown in black.

A9: Thank you. We have corrected.

Q10: Line 286 – 287: “As a result, silver ion concentrations in the media with or without the chamber were at similar levels (Figure S3).”: This statement seems incorrect. Fig. S3b shows that the silver ion concentration in the chambered condition was lower at 700 and 1400 uM SSD, as compared with non-chambered conditions.

A10: Thank you for your precise comment. From the graph in Fig. S3, we may say that SSD (direct) seems a legalistic regression, and SSD (chambered) seems a linear regression. Therefore, we have changed the sentences using your suggestion as “As a result, silver ion concentrations in the chambered condition were lower at 700 and 1400 μM SSD, as compared with non-chambered conditions, but, in higher concentrations more than 2800 μM, they became similar levels (Figure S3).

Q11: According to Ref # 32, SSD was not effective in killing MRSA in biofilm. I think this should be discussed.

A11: Thank you for your important comments. We have discussed about this in Discussion.